# Pacing Profiles of Middle-Distance Running World Records in Men and Women

**DOI:** 10.3390/ijerph182312589

**Published:** 2021-11-29

**Authors:** Arturo Casado, Fernando González-Mohíno, José María González-Ravé, Daniel Boullosa

**Affiliations:** 1Center for Sport Studies, Rey Juan Carlos University, 28933 Madrid, Spain; arturo.casado@urjc.es; 2Sport Training Laboratory, Faculty of Sport Sciences, University of Castilla la Mancha, 45004 Toledo, Spain; josemaria.gonzalez@uclm.es; 3Facultad de Ciencias de la Vida y de la Naturaleza, Universidad Nebrija, 28240 Madrid, Spain; 4Integrated Institute of Health, Federal University of Mato Grosso do Sul, Campo Grande 79070-900, Brazil; daniel.boullosa@gmail.com; 5College of Healthcare Sciences, James Cook University, Townsville 4811, Australia

**Keywords:** pacing, middle-distance running, world record, athletics

## Abstract

The aims of the current study were to compare the pacing patterns of all-time 800 m, 1500 m and mile running world records (WRs) and to determine whether differences exist between sexes, and if 800 m and 1500 m WRs were broken during championship or meet races. Overall and lap times for men and women’s 800 m, 1500 m, and mile WRs from World Athletics were collected when available and subsequently compared. A fast initial 200 m segment and a decrease in speed throughout was found during 800 m WRs. Accordingly, the first 200 m and 400 m were faster than the last 200 m and 400 m, respectively (*p* < 0.001, 0.77 ≤ ES ≤ 1.86). The first 400 m and 409 m for 1500 m and mile WRs, respectively, were faster than the second lap (*p* < 0.001, 0.74 ≤ ES ≤ 1.46). The third 400 m lap was slower than the last 300 m lap and 400 m lap for 1500 m and mile WRs, respectively (*p* < 0.001, 0.48 ≤ ES ≤ 1.09). No relevant sex-based differences in pacing strategy were found in any event. However, the first 409 m lap was faster than the last 400 m lap for men but not for women during mile WRs. Women achieved a greater % of WRs than men during championships (80% vs. 45.83% in the 800 m, and 63.63% vs. 31.58% in the 1500 m, respectively). In conclusion, positive, reverse J-shaped and U-shaped pacing profiles were used to break 800 m, men’s mile and 1500 m, and women’s mile WRs, respectively. WRs are more prone to be broken during championships by women than men.

## 1. Introduction

Pacing, described as the work or effort distribution over a race, has been extensively studied over the last 40 years in endurance sports [1]. There is consensus that pacing is a prerequisite to achieve successful endurance performance, and that depends on several internal (i.e., muscle fatigue [2], and psychophysiological variables [3]) and external (i.e., tactics [4], and ambient conditions, such as wind resistance [5]) factors [6]. Ideally, an even effort distribution would be the most optimal pacing strategy from an energetic point of view [7], but the accumulated evidence demonstrates that different pacing profiles can be observed mostly depending on the distance/duration of the race [8].

The analysis of pacing strategies of track running world records (WRs) is an excellent paradigm of study because they were used to achieve the most optimal and outstanding performances in history. These strategies may be considered by runners exhibiting lower performance and help them to improve it through a learning process. Interestingly, while cross-sectional data suggest a reduced variability of velocity with WRs improvements over different distances [7,9], a more recent analysis of WRs performed by the same athletes suggest that these individuals’ running performances can be improved without changes in their own pacing strategies [10]. Pacing strategies in men during 800 m [11], 1500 m [12] and mile [13] running WRs were previously studied. However, the analysis of men’s 800 m WRs is not updated in the current literature, with the latest three WRs achieved by Kenyan runner David Lekuta Rudisha. Therefore, considering the new WRs and the potential impact of exceptional WR holders on performance and pacing, such as Rudisha for 800 m, it is required an updated analysis of pacing strategies of middle-distance running races in which the contribution of anerobic capacity is more relevant [14] than in other endurance running races. In addition, to date, no previous study has analyzed the pacing strategies during women’s 800 m and 1500 m WRs.

Sex influence on pacing behavior is also another topic of interest in pacing research, as it would be expected that internal factors associated to sex differences may be relevant. A previous study compared both men’s and women’s pacing strategies from marathon WRs [15] and found that women tended to follow a less uniform pace, while men typically adopted a more even pace with a fast end spurt at the final stages of the race. Regarding middle-distance running WRs, Foster et al. [7] observed that the pacing pattern of men’s mile WRs is characterized by a progressive reduction in the within-lap variation of pace, while in women, the pattern of lap times has almost not changed over time, likely secondary to a lack of performance depth in the women’s fields. However, this previous analysis of mile WRs in women was conducted up to 1996. In addition, a sex-based comparison between 800 m and 1500 m WRs is warranted to better understand whether physiological differences between sexes influence pacing patterns over middle-distance runs.

The use of pacemakers strongly assist for the achievement of the fastest possible finishing performance by means of a reduction in the cognitive load associated to a continuous decision-making process [3,16], and also allowing WR aspirants to take advantage of drafting [3,5]. However, pacemakers are typically used during meets (i.e., non-championship races in which the main goal is to achieve the fastest finishing performance) rather than championship races [8]. In addition, setting a WR is a very different goal to winning a gold medal during major championships. Therefore, it is expected that middle-distance running WRs were achieved during meets rather than championship races. However, WRs may also be achieved during championship races, at which world-class athletes typically peak. Nonetheless, to date, no previous study analyzed the type of race (i.e., meets vs. championship races) in which WRs were broken.

Therefore, the aims of the current study were (1) to describe and compare the pacing profiles of all-time middle-distance running WRs, (2) to verify whether differences exist between men and women, and (3) to determine whether 800 m and 1500 m WRs were broken during either championship or meet races.

## 2. Materials and Methods

### 2.1. Pacing Data

Overall and split times recorded during 800 m (21 men and 10 women), 1500 m (37 men and 10 women) and mile (32 men and 9 women) world records (WRs) from the World Athletics (WA, formerly International Amateur Athletic Federation (IAAF)) era until 2014 were collected from the Hymans and Metrahazi [17] database when available. WRs ratification by WA and 2 other WRs broken from 2015 to 2020 were extracted from the WA website (www.worldathletics.org (accessed date 20 December 2020)).

### 2.2. Design and Methodology

The present study followed an observational approach. In the men’s 800 m event, 24 WRs were ratified by WA from 1912 to 2012. However, WRs with split times in yards or without split times were excluded. Finally, 400 m lap times were available for 13 races, while 200 m lap times were available for the remaining 8 races. They represented 87.5% of all WRs. In women’s 800 m events, 29 WRs from 1922 to 1983 were ratified by WA. However, split times were only available for 10 WRs, which represented 34.5% of all WRs. Finally, 200 m and 400 m lap times were available in 2 and 8 WRs, respectively.

In the men’s 1500 m event, 38 WRs were ratified by WA from 1912 to 1999. Lap times (3 × 400 m lap times and the last 300 m) were available for 37 men’s WRs from 1917 to 1998. They represented 97.4% of all WRs. In the women’s 1500 m event, 14 WRs were ratified by the WA from 1967 to 2015. Only split times for 11 WRs were available. They represented 78.6% of all WRs.

In the men’s mile event, 32 WRs were ratified by WA from 1913 to 1999. Lap (a first lap of 409 m and 3 × 400 m laps) times were available in all these WR. In the women’s mile event, 14 WRs from 1967 to 2019 were ratified by WA. Split times were available in 10 WRs, which represented 71.4% of all WRs.

Each lap time was expressed as a percentage of the average race speed (%RS) for further comparisons. Categorization of the different pacing strategies was conducted according to the statistical differences found between lap times. For example, a U-shaped pacing strategy during either 1500 m or mile events is considered if the first and second laps are covered at significantly faster paces than those during the second and third laps, without significant differences between paces during the first and last laps. Rather, a reverse J-shaped pacing strategy is considered if the first and last laps are covered at significantly faster paces than those during the second and third laps, and the first lap is completed at a significantly faster pace than that during the last lap.

### 2.3. Statistical Analysis

All data are presented as mean and standard deviation (mean ± SD). Data were checked for normality of distribution, equality of variances, and assumption of sphericity. When the sphericity assumption was violated, the Greenhouse–Geisser correction was employed. A 2-factor analysis of variance (ANOVA) with repeated measures with ‘race average speed at each lap’ as the between laps’ factor and sex as the between subjects’ factor was conducted to determine the differences between %RS at each lap and between sexes. A Bonferroni post hoc correction was used in all pairwise comparisons. Effect sizes (ES) were calculated using partial eta-squared (ηp2) for the repeated measured ANOVA test, and Cohen’s d [18] for the Bonferroni post hoc test. The ηp2 was considered to be small (0.01), moderate (0.01–0.06) or large (>0.15) [19]. The Cohen’s d was considered to be small (0.21–0.50), moderate (0.51–0.80) or large (>0.80) [18]. Statistical significance was set at *p* < 0.05. All analyses were performed with the JASP software (version 0.13.1 for Mac OS, JASP Team, Amsterdam, the Netherlands). Figures were performed with the Graph Pad Prism software (version 8.0 for Mac) (San Diego, CA, USA).

## 3. Results

The repeated measures ANOVA revealed a non-significant difference in %RS between sexes for all middle-distance running events (Table 1). However, there were significant differences in %RS within all middle-distance events in both sexes.

In the 800 m event (Figure 1), the first 400 m lap was covered at a significantly faster speed than the second 400 m lap (*p* < 0.001, ES = 0.99 and 0.77 for men and women, respectively). The first 200 m lap was also covered at a significantly faster speed than the fourth 200 m lap for men (*p* < 0.001, ES = 1.86).

In the 1500 m event (Figure 2), the first lap was covered at a significantly faster speed than the second lap (*p* < 0.001, ES = 0.91 and 0.78 for men and women, respectively). The second lap was covered at a significantly slower speed than the last 300 m lap (*p* < 0.001 and *p* = 0.05, ES = 1.02 and 0.65 for men and women, respectively). The third lap was covered at a significantly slower speed than the last 300 m split (*p* < 0.001, ES = 0.86 and 0.48 for men and women, respectively). There were no differences between %RS at the first lap and the last 300 m lap for men or women.

In the mile event, the first 409 m lap was covered at a significantly faster speed than the second lap (*p* < 0.001, ES = 1.46 and 0.74 for men and women, respectively), and also than the third lap (*p* < 0.001, ES = 1.62 and 0.78 for men and women, respectively). The men’s first lap was faster than the last lap (*p* < 0.05, ES = 0.53), the second lap was slower than the last lap (*p* < 0.001, ES = 0.93), and the third lap was slower than the last lap (*p* < 0.001, ES = 1.09) (Figure 3).

Championship races were used for the achievement of 45.83% and 80%, and 31.58% and 63.63%, of men’s and women’s 800 m and 1500 m WRs, respectively. The remaining ones were set during meet races.

## 4. Discussion

The aims of the current study were to describe and compare the pacing patterns of all-time 800 m, 1500 m and mile running WRs to determine if differences exist between sexes and identify whether 800 m and 1500 m WRs were broken during either championship or meet races. Given that pacing patterns in women’s 800 m and 1500 m WRs were not previously studied to the best of the authors’ knowledge, the most important finding is that a very similar pacing approach was followed in these events when compared to men. However, pacing strategies in the mile event differed among sexes. In this way, both men and women’s 800 m WRs displayed a positive pacing profile (i.e., characterized by a decreasing speed throughout the race), while both men and women’s 1500 m, and women’s mile WRs were characterized by a U-shaped pacing profile (i.e., displaying a faster–slower–slower–faster pattern for each successive lap). Finally, men’s mile WRs followed a reverse-J-shaped pacing profile characterized by a very fast start, a slower middle part of the race, and an end-spurt, which was found to be slower than the first lap of the race.

The positive pacing profile observed in both men and women’s 800 m WRs is consistent with findings from a previous study which examined the men’s 800 m WRs set until 1997 [11]. This pacing profile was found to be the same after including the three WRs set by Rudisha. Moreover, the current men’s WR was characterized by an extremely positive pacing profile (i.e., the speed decreases throughout the race) in which the first 200 m and 400 m splits were covered in 23.4 s and 49.28 s, respectively, with a final lap of 51.63 s (Figure 1B) [14,17]. This pattern observed during 800 m WRs can be also considered a seahorse-shaped [20] pacing profile (see Figure 1). Similarly, this seahorse-shaped profile was also observed during the qualifying rounds and final races at the 2013 and 2017 WA Championships and 2016 Olympic Games in which the fastest speed was achieved over 200 m, followed by a deceleration to 300 m, a constant speed to 500 m, another acceleration to 600 m, and then maintenance or deceleration up to the finish line [20]. Moreover, no substantial differences seem to exist between the pacing profile used to achieve WRs and that used to succeed during global championship races in the 800 m event. In this regard, a more positive strategy in elite and world-class speed-based milers than endurance-based milers was observed not only during competitions, but also during their training practices [21]. This would suggest an influence of training background on pacing patterns that warrants further research.

The lack of sex-based differences in pacing profiles found in 800 m WRs is not consistent with results from a previous study analyzing 142 of the 800 m season best performances of world-class athletes during meet races [22]. This previous study found that the first 200 m was always the fastest. However, whereas men tended to decrease the pace progressively throughout the race, there were no pace differences between the last three 200 m laps for women. In this way, men covered the second 200 m segment at a relatively faster speed than women. This sex-based difference was attributed to the lower performance level displayed during women’s races, which may have influenced their relatively slower second 200 m segment [22]. Nonetheless, these differences in performance level do not seem to affect pacing profiles during 800 m WRs, as they represent the greatest performances achieved ever for both men and women. However, an important sex-based difference was found regarding the greater number of WRs set during championship races by women than men (i.e., 80% vs. 45.83% of the WRs analyzed for women and men, respectively). This finding emphasizes the competitiveness of the 800 m WRs set by women, as, during championship races, the use of pacemakers is not allowed. Of note, 800 m major championships involve participation in different qualifying rounds within just a few days in order to be able to run the final race, which, in turn, leads to high levels of physical fatigue, thus preventing runners from being in the same resting state to that typically experienced during meet races.

Similar to findings from previous studies analyzing men’s 1500 m [12] and mile [13] WRs broken until 1999 and 1998, respectively, a U-shaped pacing strategy was found on average during both men and women’s 1500 m and women’s mile WRs. However, contrary to the results reported by Noakes et al. [13], a reverse-J-shaped rather than U-shaped pacing strategy was found for the latest 32 men’s WRs. These represent a quite different pacing profile to that observed in the 800 m event and highlights a shift from a positive/seahorse-shaped strategy toward a U-shaped pacing strategy as long as the event distance increases. In the 800 m event, the initial speed clearly exceeds the fatigue threshold running speed, i.e., the fastest speed that can be maintained throughout any entire event, which is determined by physiological limits, such as pulmonary gas exchange, blood lactate concentrations, and acid base status [23]. In this manner, 800 m runners breaking WRs typically reach their metabolic limit to compensate for the lost time later in the race [11].

However, during the 1500 m and mile WRs, the fatigue threshold is not surpassed to that extent, as milers are able to display a fast end spurt in the latest stages of the race, showing a greater ability than 800 m runners to recover from the first ‘fast’ 400 m lap [21]. Furthermore, the progression in pacing profiles of men’s mile WRs is characterized by a shift from U-shaped toward a more even pacing strategy [7], showing, therefore, the high importance of not surpassing early the fatigue-threshold speed in order to optimize performance in these middle-distance running events. By contrast to the 800 m event, the pacing profiles obtained in both men and women 1500 m WRs substantially differed from those observed during the qualifying rounds and final races at the 2013 and 2017 WA Championships and 2016 Olympic Games, which were characterized by a J-shaped pacing profile [20]. This pacing profile involves important variations in pace during the race [24] in which runners adopt a moderate initial speed during the first lap, decelerate the speed for the second lap, increase the speed between 700 m and 1300 m, and maintain or decelerate their speed during the final 200 m [20]. Indeed, success during the men’s 1500 m qualifying rounds of the 2017 WA Championships was related to being able to generate a fast end spurt during the last 300 m of the race [25]. In this manner, during championship races in which the main goal is achieving either a qualifying position to participate in the next round or the highest possible position during the final race, the end spurt is typically faster and the middle part of the race is slower than during races in which WRs are achieved [24]. However, despite a U-shaped pacing profile being found for all 1500 m and mile WRs, both current men’s 1500 m and mile WRs set by Moroccan Hicham El Guerrouj displayed a J-shaped pacing profile (Figure 2B and Figure 3B), similar to that observed during championship races [20]. In addition, contrary to the findings by Noakes et al. [13], who concluded that no pace differences were found between the first 409 m lap and last 400 m lap for the latest 32 men’s mile WRs, we observed that the first lap was faster than the last lap, thus revealing a reverse-J-shaped pacing strategy. These different results in the pacing strategy used for the same WRs can be explained by the different type of analysis conducted between studies. While Noakes et al. [13] calculated the differences between the average lap time, we analyzed the differences between the average lap speed, which is more appropriate for relative comparisons.

Similar to findings in the 800 m event, no sex-based differences were found in 1500 m and mile WRs. Nonetheless, a reverse-J-shaped pacing strategy in which the first 409 m lap was faster than the last 400 m lap was adopted in men’s mile WRs, being different from the U-shaped profile observed for women. Furthermore, also similar to the findings in the 800 m event, a greater % of women’s 1500 m WRs were achieved during championship races (i.e., 63.63% vs. 31.58% of the WRs analyzed for women and men, respectively). These sex-based differences found are in agreement with findings from several studies, indicating that women are better pacers than men, as they tend to slow less during the latest stages of long-distance races, such as 10 km road races [26], cross-country races [27] and marathons [28,29]. Rather than attributing the greater % of women’s WRs achieved during championship races to sex-related physiological characteristics, such as those proposed for longer distance events (i.e., a greater amount of type I muscle fibers in women than men [30]), the most likely explanation may be the lower density in performance level in women’s races, which may have allowed to set faster initial paces without a serious threat of being defeated by a competitor of similar ability [8]. In addition, a greater number of WRs were beaten in championship races during the 800 m rather than the 1500 m events. Given that the use of pacemakers is scarce in championship races, these results suggest that the longer the distance of the event, the more important the use of pacemakers in order to achieve a fastest performance. This finding clearly emphasizes the high mental cost of the decision-making process represented by continuously setting the pace during endurance events, apart from the benefits derived from drafting [3,16].

The lack of both official electronic splits for some of the WRs analyzed and the exclusion of a substantial amount of WRs, especially in women events, have to be acknowledged as the main limitations of the present study. In this way, outcomes derived from the statistical comparisons between lap times during women’s mile WRs may be affected to some extent by the reduced sample size already available.

## 5. Conclusions

A positive pacing strategy characterized by a fast 200 m, a deceleration to 600 m, and another deceleration to the finish line was found during 800 m WRs. A U-shaped pacing strategy was found on average during 1500 m and women’s mile WRs. A reverse-J-shaped pacing strategy characterized by a fastest–slower–slower–faster pattern was followed during men’s mile WRs. No relevant sex-based differences in pacing strategy were found in any of these three middle-distance running events. Women achieved a greater % of 800 m WRs than men during championship races. Similarly, women also set a greater % of 1500 m WRs than men during championship races. Therefore, as long as the distance of the event increases, the number of WRs broken at championship races decreases.

### Practical Applications

Men and women’s middle-distance runners aiming to break the 800 m WR should cover a fast initial 200 m segment of the race while trying to lose the least possible speed during the rest of the race. On the other hand, men and women runners attempting to achieve the 1500 m and mile WRs may adopt either a U-shaped pacing profile characterized by a faster–slower–slower–faster pattern for each successive lap, or an even pacing strategy in which the fatigue threshold speed should not be surpassed. However, a more even pacing strategy may be likely more effective for doing so [7]. Women’s 800 m runners peaking for a major championship may consider breaking a WR while chasing the gold medal in the case that the pre-expected performance difference with their competitors is substantial. This pre-expectation can be considered according to the differences between the athletes’ season best times and those from the closest rivals, and through the information obtained from the performance achieved during the last competitions prior to the championship. For men, in the case that performance differences between the theoretically best male 800 m runner and the rest of the runners are very large, such as that exhibited by David Rudisha and his rivals at the 2012 London Olympic Games, a WR attempt may be also achieved.

## Figures and Tables

**Figure 1 ijerph-18-12589-f001:**
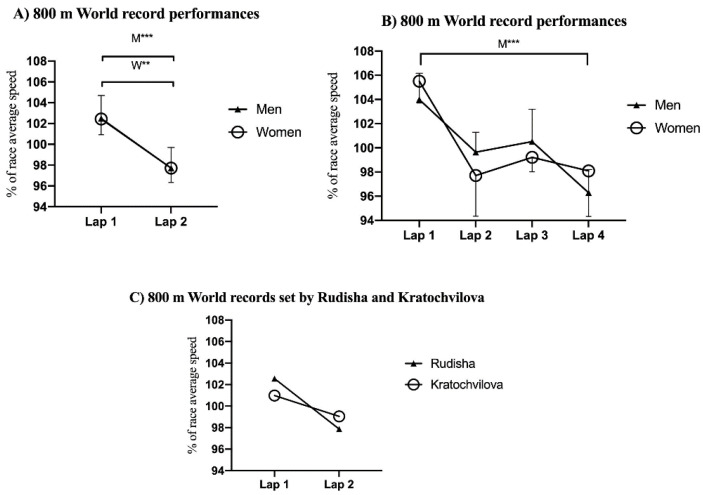
Mean and standard deviation of (**A**) average race speed of two 400 m laps, (**B**) four 200 m laps during male and female 800 m world record performances, and (**C**) world records set by Rudisha and Kratochvilova with two 400 m laps. ** *p* < 0.01; *** *p* < 0.001.

**Figure 2 ijerph-18-12589-f002:**
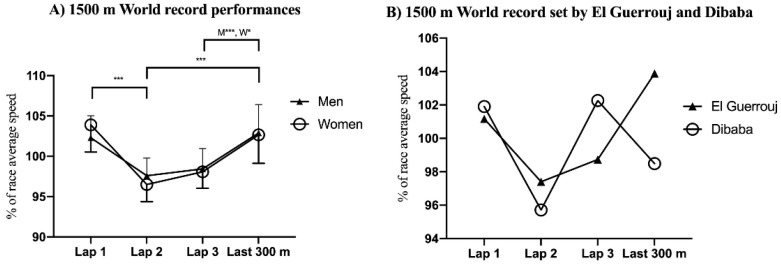
Mean and standard deviation of (**A**) average race speed of three 400 m laps and last 300 m during men and women’s 1500 m world record performances (**A**), and (**B**) world records set by El Guerrouj and Dibaba. * *p* < 0.05; *** *p* < 0.001.

**Figure 3 ijerph-18-12589-f003:**
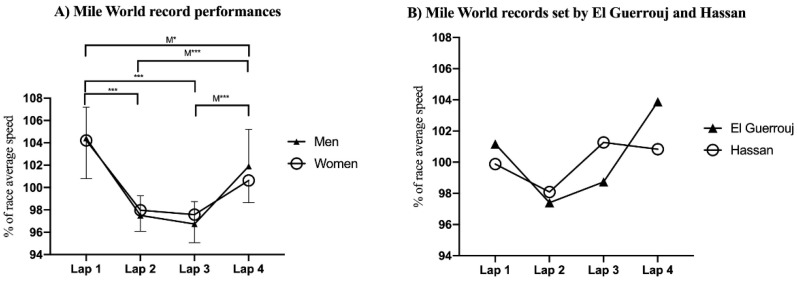
Means and standard deviation of (**A**) average race speed of first 409 m lap, and three 400 m laps during men’s and women’s mile world record performances, and (**B**) world records set by El Guerrouj and Hassan. * *p* < 0.05; *** *p* < 0.001.

**Table 1 ijerph-18-12589-t001:** Repeated measures ANOVA outcomes of lap and lap and sex interaction.

			Repeated Measures ANOVA						
			Between Laps				Lap and Sex Interaction		
Event	Number of Laps	df Residual	*p*	F	df	ES	*p*	F	ES
800 m	2	19	<0.001	31.55	1	0.623	0.978	0.00	0.000
	4	21	<0.001	10.029	3	0.561	0.483	0.483	0.047
1500 m	4	138	<0.001	31.40	3	0.397	0.358	1.08	0.014
Mile	4	95.80	<0.001	39.54	2.39	0.492	0.486	0.771	0.010

F—variation between sample means/variation within the sample; df—degrees of freedom; ES—effect size (ηp^2^).

## Data Availability

Not applicable.

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
