# Peer review of "Pacing Profiles of Middle-Distance Running World Records in Men and Women"

_ijerph, 2021, doi:10.3390/ijerph182312589_

Round 1

Reviewer 1 Report

Dear Authors,

I read with interest your manuscript, which I found logical and fairly well-written. 

I just have a few comments, exposed below:

Keywords: I guess it is refuse reporting "Pacing-middle" and "distance running". Maybe you meant to write "Pacing" and "Middle-distance running".

Introduction: I do not find useful in the context of the paragraph the sentence at lines 43-44, starting with "These strategies..." and ending with "...pacing abilities". Consider removing.

Introduction, Line 49: "[1+3" I think is an error, and it should be "[13]" as the number of 13th reference number.

Statistical analysis

- If partial eta-square was used, please add the "p" letter in subscript ( ).

Results

- When reporting ANOVA results in Table 1, please indicate also the Degrees of Freedom for the error, not only the df of the conditions (laps in your case). 

Figures

In Figure 1A, are Men data missing? Or they are just covered by the Women data? Seems unlikely the latter, 'cause also error bars should be exactly the same.

Further, since MDPI should not add additional costs for colour figures, why don't you consider using different colours for male and female data?

You might also consider using the same y-scale for all the plots, in order to allow an easier comparison among them. 

Lastly, in plots 2B and 3B, circle and square are reversed (with a square for the male and a circle for the female athlete) respect all the others. Please, be consistent with the use of the symbols.

Discussion, Lines 184-187: Please check the correctness of this sentence, as I find it more correct in the singular form, as "current men’s WR was characterized by an extremely positive pacing profile (i.e., the speed decreases throughout the race) in which the first 200-m and 400-m splits were covered in 23.4s and 49.28s, respectively, with a final lap of 51.63s". 

Conclusion: Please remove the data within parenthesis in lines 290-293, as they have been already reported in the results and discussion sections. 

Other comments

Sincerely, I found a bit tricky to infer the difference between U-shape and reverse-J-shape profiles. Now - if I am not wrong - both share a pattern "fast-slow-slow-fast", but in the U-shape the latter is not different from the first, while in the J-shape, the latter section is slower than the first lap. I believe you should give a more precise definition and explanation of the two profiles, and how do you categorise the race in one or the other pattern. Then, if you categorise the patterns in U or J based on the results of the ANOVA and post-hoc analyses, in my opinion, these conclusions are strongly affected by n (for example, for the mile race is 32 for males and 10 for females). My point is: is it possible that you categorised as J-shape races which maybe are instead U-shape, but a significant difference is not present due to the small sample size? If you agree, maybe this point deserves to be discussed or acknowledged in the limitations.

To conclude, but this is just a curiosity, did you check or find any difference in the pacing strategies between years? I mean, as you state in lines 233-236, "the progression in pacing profiles of men’s mile WRs is characterized by a 233 shift from U-shaped towards a more even pacing strategy", but this is supported by your data? Are the pacing strategies really changing, maybe led by a more scientific approach?

I look forward to receiving your answers to my comments.

Kind regards.

Author Response

Reviewer 1,

Dear Authors,

I read with interest your manuscript, which I found logical and fairly well-written. 

I just have a few comments, exposed below:

Keywords: I guess it is refuse reporting "Pacing-middle" and "distance running". Maybe you meant to write "Pacing" and "Middle-distance running".

Thanks for this comment. We have now corrected it as suggested.

Introduction: I do not find useful in the context of the paragraph the sentence at lines 43-44, starting with "These strategies..." and ending with "...pacing abilities". Consider removing.

Thanks for this comment. We have now substituted that for the following sentence in L45: “These strategies may be considered by runners exhibiting lower performance and help them to improve it through a learning process”.

Introduction, Line 49: "[1+3" I think is an error, and it should be "[13]" as the number of 13th reference number.

Thanks for your appreciation, which is correct. We have now addressed it.

Statistical analysis

- If partial eta-square was used, please add the "p" letter in subscript ( ). 

Thanks for your comment. We have now included the “p” letter.

Results

- When reporting ANOVA results in Table 1, please indicate also the Degrees of Freedom for the error, not only the df of the conditions (laps in your case). 

We have now included it in the Table 1.

Figures

In Figure 1A, are Men data missing? Or they are just covered by the Women data? Seems unlikely the latter, 'cause also error bars should be exactly the same. 

The % of race average speed were the following:

Men Lap 1: 102.49 ± 2.20

Men Lap 2: 97.71 ± 1.99

Women Lap 1: 102.44 ± 1.52

Women Lap 2: 97.71 ± 1.38

For that reason, the line and symbols in the graph are overlapping. We have changed the symbols for a better understanding. 

Further, since MDPI should not add additional costs for colour figures, why don't you consider using different colours for male and female data?

Thanks for your suggestion. We have now modified the symbols for a better understanding.

You might also consider using the same y-scale for all the plots, in order to allow an easier comparison among them. 

We have now modified the y-scale for comparison among them.

Lastly, in plots 2B and 3B, circle and square are reversed (with a square for the male and a circle for the female athlete) respect all the others. Please, be consistent with the use of the symbols.

Thanks for this comment. We have now included the same symbols across all figures.

Discussion, Lines 184-187: Please check the correctness of this sentence, as I find it more correct in the singular form, as "current men’s WR was characterized by an extremely positive pacing profile (i.e., the speed decreases throughout the race) in which the first 200-m and 400-m splits were covered in 23.4s and 49.28s, respectively, with a final lap of 51.63s". 

Thanks for this comment. We have now corrected it as suggested.

Conclusion: Please remove the data within parenthesis in lines 290-293, as they have been already reported in the results and discussion sections. 

Thank you for your appreciation. We have removed it.

Other comments

Sincerely, I found a bit tricky to infer the difference between U-shape and reverse-J-shape profiles. Now - if I am not wrong - both share a pattern "fast-slow-slow-fast", but in the U-shape the latter is not different from the first, while in the J-shape, the latter section is slower than the first lap. I believe you should give a more precise definition and explanation of the two profiles, and how do you categorise the race in one or the other pattern. Then, if you categorise the patterns in U or J based on the results of the ANOVA and post-hoc analyses, in my opinion, these conclusions are strongly affected by n (for example, for the mile race is 32 for males and 10 for females). My point is: is it possible that you categorised as J-shape races which maybe are instead U-shape, but a significant difference is not present due to the small sample size? If you agree, maybe this point deserves to be discussed or acknowledged in the limitations.

Thanks for this comment. We agree with it and, therefore, we have now explained the specific categorization criterium used based on the statistical analyses conducted as follows in L113: “Categorization of the different pacing strategies was conducted according to the statistical differences found between lap times. For example, a U-shaped pacing strategy during either 1500m or mile events is considered if the first and second laps were covered at significantly faster paces than those during the second and third laps, without significant differences between paces during the first and last laps though. Rather, a reverse J-shaped pacing strategy is considered if the first and last laps were covered at significantly faster paces than those during the second and third laps, and the first lap was completed at a significantly faster pace than that during the last lap”. In addition, we have now indicated that the reduced amount of women’s mile WRs available may have affected the outcomes of the statistical analyses conducted in L316 as follows: “In this way, outcomes derived from the statistical comparisons between lap times during women’s mile WRs may be affected to some extent by the reduced sample size already available”.

To conclude, but this is just a curiosity, did you check or find any difference in the pacing strategies between years? I mean, as you state in lines 233-236, "the progression in pacing profiles of men’s mile WRs is characterized by a 233 shift from U-shaped towards a more even pacing strategy", but this is supported by your data? Are the pacing strategies really changing, maybe led by a more scientific approach?

I look forward to receiving your answers to my comments.

Kind regards.

Thanks for this comment. However, it was not the aim of this study given the absence of sufficient amount of data to conduct properly this type of analysis. In any case, we consider that it is a good idea for a further study which may only focus on the events in which a sufficient amount of WRs is available to properly conduct this type of analysis.

Reviewer 2 Report

Dear authors, thank you for your paper. I found it very interesting. Most of my comments are in my PDF file. I feel the the English is good, in general the paper is well written. I come from an athletics background, but not from middle distance. My big question is if you analyze the laps from the 1500m start (300m mark on track, 700m, 1100m) is this practical for the athletes? At some events there may be timers giving feedback here, but I am assuming that this does not happen at big races. Comments? My other comments are in the PDF. If you have the time, I would like to discuss this paper further, so please try to respond to my comments in the PDF, and perhaps we can continue this discussion through emails at some point. Thank you for an interesting paper!

Author Response

Reviewer 2

Page 2

Line 101: ¿?

Sorry about the mistake. It is the reference number 13.

Page 3

Line 102: I assume that you mean at 700, 1100? The first relay zone. The start of the 1500m. If so, do athletes get split times here? Or do they just see the clock at the finish line? I am just wondering if analyzing here makes sense if the athlete gets no feedback at this point in the race. I THINK that in big meets the only feedback is at 400, 800, 1200 (finish line) and the big screen. Do you understand what I am trying to say?

In the 1500-m event, the lap times were: first 400m (from the beginning of the race), second lap 400-800m, third lap from 800-1200m, and the fourth lap from the 1200m-finish line.

Page 4

Line 134: Sorry - maybe I missed somehing here - there is not line for men in the first (A) figure.

The % of race average speed were the following:

Men Lap 1: 102.49 ± 2.20

Men Lap 2: 97.71 ± 1.99

Women Lap 1: 102.44 ± 1.52

Women Lap 2: 97.71 ± 1.38

For that reason, the line and symbols in the graph are overlapping. We have changed the symbols for a better understanding. 

Page 6

Line 202: I don't thnk that you need "whereas" here. Either "but" OR "whereas" but not both.

Thanks for this comment. We have now changed the full sentence as it was too long. The following sentences in L233-237 are the current ones: “This previous study found that the first 200-m was always the fastest. However, whereas men tended to decrease the pace progressively throughout the race, there were not pace differences between the last three 200-m laps in women. In this way, men covered the second 200-m segment at a relatively faster speed than women”.

Line 220: Am I misunderstanding something here? Both Hassan and Dibaba were slower in laps 2 and 4 - this is no U shape. Lap 3 faster than 2 but then lap 4 slower than 3 again. Not a U. Please address this. If you mean the AVERAGE trend in 1500m and mile for the women is U shaped, then please change the wording of this sentence.

Thanks for this comment. We have now modified the sentence in L251 as follows: “…a U-shaped pacing strategy was found on average during both men’s and women’s 1500-m and women’s mile WRs”.

Pag 7

Line 274-279: I find this confusing. How often are pacemakers used at championship events? How many countries have enough depth to have athletes sacricfice their result at a championship to act as a pacemaker for their countryman? Please address this. I am a bit confused here.

Thanks for this comment. We already meant what the reviewer has indicated. Therefore, we have now clarified this sentence in L308 as follows: “Given that the use of pacemakers is scarce in championship races, these results suggest that the longer the distance of the event, the more important is the use of pacemakers in order to achieve a fastest performance”.

Pag 7

Line 285: I disagree. According to the figures of Hassan's and Dibaba's WRs, these are not U shapes.

Thanks for this comment. We have now reworded the sentence in L322 as follows: “A U-shaped pacing strategy was found on average during 1,500-m and women’s mile WRs”.

Line 297: Is "speed reserve" a concept that everyone knows and understands? THis may be a concept that some readers are not familiar with.

Thanks for this comment. Therefore, we have now removed part of the sentence in L333 and it remains as follows: “Men’s and women’s middle-distance runners aiming to break the 800-m WR should cover a fast initial 200-m segment of the race while trying to lose the least possible speed during the rest of the race”.
